# Increased Epidermal Nerve Growth Factor without Small-Fiber Neuropathy in Dermatomyositis

**DOI:** 10.3390/ijms23169030

**Published:** 2022-08-12

**Authors:** Lai-San Wong, Chih-Hung Lee, Yu-Ta Yen

**Affiliations:** 1Department of Dermatology, Kaohsiung Chang Gung Memorial Hospital and Chang Gung University College of Medicine, Kaohsiung 833, Taiwan; 2Department of Dermatology, Fooyin University Hospital, Pingtung 928, Taiwan; 3Institute of Biomedical Sciences, National Sun Yat-sen University, Kaohsiung 804, Taiwan

**Keywords:** cutaneous lupus erythematosus, dermatomyositis, nerve growth factor, small-fiber neuropathy

## Abstract

Small-fiber neuropathy (SFN) is suggested to be involved in the pathogenesis of some types of autoimmune connective tissue diseases. SFN with a reduction in epidermal nerve fibers might affect sensory fibers and cause neuropathic symptoms, such as pruritus and pain, which are common in both dermatomyositis (DM) and cutaneous lupus erythematosus (CLE). Nerve growth factor (NGF) has been recognized as important in nociception by regulating epidermal nerve fiber density and sensitizing the peripheral nervous system. The present study aimed to investigate whether SFN was associated with the cutaneous manifestations of DM and CLE. We also investigated the relationship between SFN and axon guidance molecules, such as NGF, amphiregulin (AREG), and semaphorin (Sema3A) in DM and CLE. To explore the molecular signaling, interleukin (IL)-18 and IL-31, which have been implicated in the cutaneous manifestation and neuropathic symptoms in DM, were examined in keratinocytes. Our results revealed that intraepidermal nerve fiber density (IENFD) was unchanged in patients with DM, but significantly reduced in IENFD in patients with CLE compared with healthy control. Increased epidermal expression of NGF and decreased expression of Sema3A were demonstrated in patients with DM. Furthermore, IL-18 and IL-31 both induced the production of NGF from keratinocytes. Taken together, IL-18 and IL-31 mediated epidermal NGF expression might contribute to the cutaneous neuropathic symptoms in DM, while SFN might be important for CLE.

## 1. Introduction

Small-fiber neuropathy (SFN), caused by the damage to cutaneous Aδ and C fibers, may be involved in the pathogenesis of some types of autoimmune connective tissue diseases. It might affect sensory fibers and cause neuropathic symptoms, such as pruritus and pain [1]. Pruritus accompanied by pain and a burning sensation is frequently noted, and develops in up to two-thirds of patients with dermatomyositis (DM) in different populations [2]. Meanwhile, pruritus is also the most common initial symptom in DM [2], and could be severe enough to interfere with the quality of life [3]. In contrast with DM, pruritus was considered infrequent and mild in patients with cutaneous lupus erythematosus (CLE). However, Samotij et al. reported pruritus was experienced in 75% out of 567 patients with CLE [4] and 76.8% out of 114 CLE patients [5] in two cross-sectional studies. Furthermore, the severity was moderate to severe in more than half of the patients [5]. This indicates that cutaneous neuropathic symptoms are prominent both in patients with DM and CLE.

Alternation of epidermal innervation has been implicated in a number of skin inflammatory disorders. Nevertheless, the relevance of epidermal nerve fiber density and pruritus transduction is varied in different etiologies. Atopic dermatitis (AD), a prototypical skin inflammatory disorder, is characterized by hyperinnervation of sensory nerve fibers [6]. It is thought that alternation of the skin innervation is related to an imbalance between the axon guidance molecules, nerve elongation factors, such as nerve growth factor (NGF) and amphiregulin (AREG), and nerve repulsive factors, such as semaphorin 3A (Sema3A) in AD [7]. On the contrary, SFN with a decrease in intraepidermal nerve fiber density (IENFD) was shown in prurigo nodularis, which has featured such as intractably itchy hyperkeratotic nodules [8]. SFN is characterized in patients with systemic LE (SLE) even in non-lesional skin [9]. Nevertheless, there is limited study about the cutaneous nerve fiber density in DM. One case report showed a decrease in epidermal nerve fiber density in a DM patient with a treatment-refractory itchy scalp, suggesting SFN was involved in DM [10]. However, there was no large study evaluating the role of SFN and the pathogenic molecules in patients with DM.

Nerve growth factor (NGF), a neurotrophin, is involved in a variety of inflammatory skin disorders, such as AD and prurigo nodularis, which usually present with pruritus and hyperalgesia [11]. In addition to neuronal cell development and survival, NGF contributes to peripheral neuronal sensitization [12] and promotes inflammation through activation of multiple immune cells, which might also indirectly modulate nociception [13]. It suggests the possible role of NGF in the neuropathic symptoms of DM and CLE.

Inflammatory cytokines, such as interleukin (IL)-18 and IL-31, have been implicated in the pathophysiology of DM. IL-18 was increased in the serum and associated with disease activities and pulmonary complications in patients with DM [14]. In addition, one study demonstrated that keratinocytes-derived IL-18 has the potential to distinguish DM from CLE, since the cutaneous rash of DM and CLE is sometimes similar [15]. IL-31, a T helper 2 inflammatory cytokine and pruritogen [11], has been shown to be involved in DM-associated pruritus, with an upregulation of IL-31 in lesioned skin of DM patients with pruritus [16].

In this study, we aimed to investigate whether SNF with reduced IENFD is associated with DM and CLE. Additionally, we intended to investigate the expression of NGF in the skin in DM and CLE and its relation to IL-18 and IL-31. We had a retrospective chart review and examined the skin samples from patients with DM and CLE for analysis and had the preliminary preclinical results. We also performed in vitro study with primary human keratinocytes to investigate the molecular signaling.

## 2. Results

### 2.1. Patients Characteristics

Among the DM subjects (17 male, 18 female, 61.9 ± 13.6 years), symptoms of pruritus, pain, or burning sensation were recorded in 60% of patients. Of the CLE subjects (9 male, 20 female, 46.6 ± 14.5 years), the symptoms above were recorded in 31% of patients. For further demographic data, see Appendix A.

### 2.2. A Significant Reduction in IENFD in Patients with CLE, but No Change in IENFD in Patients with DM

We asked whether epidermal innervation would be different in patients with DM and CLE compared to healthy controls. To address this, we measured IENFD using an immunofluorescence exam (Figure 1a). Quantitative analysis showed significantly low IENFD in patients with CLE compared with healthy controls and patients with DM (*p* < 0.01) (Figure 1b). There was comparable IENFD in patients with DM compared with healthy controls (Figure 1b).

We then sub-analyzed patients with CLE and found a significant reduction in the group with cutaneous symptoms of pruritus, pain, or burning sensation (*n* = 9) in comparison with the asymptomatic group (*n* = 20) (*p* < 0.05) (Figure 1c).

We also analyzed the IENFD in different anatomic locations in patients with DM and CLE, and healthy controls. A significant reduction in IENFD was noted in different anatomic locations (*p* < 0.01) (Table 1). The IENFD in healthy control was 47.1 +/− 21.3, 34.2 +/− 9.2, 34.7 +/− 7.7 on the face, trunk, and limbs, respectively (Table 1). There was a trend of higher IENFD in the skin of the face, but it was not significant.

### 2.3. Upregulation of Epidermal NGF with No Change of AREG in Patients with DM

We have demonstrated that IENFD did not change in the skin of DM while SFN was noted in the skin of CLE. We then asked whether NGF and AREG expression in the skin might be different in patients with DM and CLE. We measured the expression of NGF and AREG with an immunofluorescence exam (Figure 2A and Figure 3A). Quantitatively, fluorescence intensity per unit area of epidermal NGF and AREG was calculated in each group, and statistical analysis was performed. The results showed expression levels of epidermal NGF were significantly increased in patients with DM compared with healthy controls and patients with CLE (*p* < 0.01) (Figure 2B). For AREG, the expression levels of epidermal AREG were similar between patients with DM and CLE, and healthy controls (Figure 3B).

### 2.4. Reduced Expression of Epidermal Sema3A in Patients with DM

Next, we checked the expression of nerve repulsive factor, Sema3A, via an immunofluorescence exam. Quantitatively, fluorescence intensity per unit area of epidermal Sema3A was calculated in each group, and statistical analysis was performed. The results showed Sema3A was significantly decreased in patients with DM compared with the healthy controls and patients with CLE (*p* < 0.01) (Figure 4).

### 2.5. IL-18 and IL-31 Both Induced NGF Expression in Cultured Keratinocytes

We further addressed the mechanism of elevated expression of NGF in the epidermis of patients with DM. Considering the aforementioned roles of IL-18 [15] and IL-31 [16] in the skin of patients with DM, we investigated the effect of IL-18 and IL-31 on NGF expression in keratinocytes. We treated the cultured primary keratinocytes with IL-18 at 5 ng/mL and IL-31 at 300 ng/mL, respectively. NGF expression was measured by Western blot. The result showed that IL-18 and IL-31induced the highest expression of NGF at 18 h (Figure 5).

## 3. Discussion

Inconsistent results about IENFD in different anatomic locations in healthy human skin were reported in previous studies. One study showed that the highest nerve fiber density was in the skin of the arm and the lowest density was in the skin of the back [17], while another group showed the highest nerve fiber density was in the skin of the back [18]. We demonstrated a trend of enhanced IENFD in facial skin, but it was not significant. The discrepancy may be due to different techniques, the age group of recruitment, and the methods of counting the IENFD.

Diabetes mellitus, familial amyloid polyneuropathy, and fibromyalgia are well-recognized disorders that contribute to SFN [19]. Previous studies also showed that SFN with a reduction in IENFD was found in patients with SLE [9,20]. The pathophysiology of SFN may differ in the different etiologies. It is known that central and peripheral nervous systems can be damaged in SLE, and peripheral neuropathy with impaired sensory perception is not rare in patients with LE [21]. Immunoglobulin and inflammatory processes in LE may damage the small nerve fibers in the skin, which might be subclinical. We demonstrated that SFN was significant in patients with CLE, but was not confined to patients with SLE (Appendix A). An association between reduced IENFD in CLE lesions and cutaneous neuropathic symptoms in our study indicates the relation between SFN and cutaneous disease activity in CLE.

On the contrary, we showed no change in IENFD in patients with DM compared with healthy controls. A case report demonstrated that decreased density of epidermal nerve fibers in the scalp in a patient with DM, who has suffered from treatment-resistant pruritus [10]. This inconsistent finding might be caused by disease duration, the influence of previous treatments and persistent scratching, and the anatomic location. Importantly, mechanisms other than SFN, such as peripheral sensitization, might take part in the cutaneous neuropathic symptoms in DM.

Keratinocytes are the major source of NGF in the skin [22], and NGF is widely known as a neurotrophic molecule that induces nerve fibers sprouting [23]. NGF production in the skin contributes to pruritus and allokinesis in patients with AD via hyperinnervation and peripheral sensitization [6]. NGF is also significantly expressed in numerous inflammatory skin disorders, such as psoriasis and prurigo nodularis [24,25]. Despite the correlation between NGF-induced hyperinnervation and skin sensitization, Hirth et al. showed NGF can primarily sensitize nociceptors without increased IENFD [26]. It has been shown that NGF can induce sensory sensitization and axonal hyperexcitability by lowering the excitatory threshold and facilitating action potential generation and conduction [12]. Recent studies illustrated that microinjection of NGF can sensitize nociceptors causing local hyperalgesia [12] and sensitize the skin to pruritus perception [27]. Therefore, our finding suggests that elevation of epidermal NGF in the lesional skin of patients with DM might sensitize peripheral nerves in the skin without influencing the numbers of epidermal nerve fibers.

Sema3A is a repulsive axon guidance molecule [28] and immunoregulator [29]. It has been demonstrated that Sema3A participates in the pathogenesis of LE [30,31,32] and autoimmune arthritis [33] by modulating B and T cell activities. The expression of Sema3A and its receptor, neuropilin-1, was decreased in peripheral blood of mononuclear cells (PBMC) in patients with SLE [31], while the expression of Sema3A was strong in the tubules in lupus glomerulonephritis [32]. However, there is comparable expression of Sema3A in CLE skin in our study. This indicates that the functional roles of Sema3A in various tissues may differ. On the other hand, Sema3A can inhibit NGF-induced sprouting of sensory afferents in the adult rat spinal cord [34] and the balance of Sema3A and NGF signaling affect the axonal growth [35]. Sema3A together with NGF acts as axon-guidance molecules in the skin in inflammatory skin disorders, such as atopic dermatitis [30,36]. However, one recent study illustrated that Sema3A attenuated neuropathic pain independent of nerve sprouting [37], and another group revealed that disruption of the Sema3A pathway did not influence motor axon regeneration [38]. This suggests Sema3A has a remodeling effect on the nervous system without changes in the density of nerve fibers. Since the role of Sema3A in DM is sparse [30], our finding of a reduction in Sema3A in keratinocytes in patients of DM indicates the possible role of imbalance of NGF and Sema3A on the neuropathic skin symptoms in patients with DM. Further investigation of the immunomodulatory function of Sema3A in the pathogenesis of DM is required.

IL-18, a member of the IL-1 family, is involved in autoimmune and skin inflammatory disorders [39,40]. We found that IL-18 enhanced the production of NGF from keratinocytes and the distinct role of IL-18 in DM-related skin presentation [15], indicating that epidermal NGF might be a crucial factor when it comes to distinguishing between DM and CLE. On the other hand, IL-31 is a well-recognized pruritogen with a late itch response when administered intradermally [41]. This suggests IL-31 might evoke itch indirectly via secondary mediators. IL-31 can modulate inflammatory response via its receptor heterodimer IL-31 receptor A/oncostatin M receptor β on keratinocytes, sensory neurons, and immune cells [11]. Here, we showed IL-31 induced NGF production from keratinocytes. This indicates IL-31- NGF pathway in the skin might be responsible for the cutaneous neuropathic symptoms in patients with DM.

In summary, our study showed that distinct mechanisms might govern the cutaneous neuropathic symptoms in DM and CLE. SFN might be important for CLE, while hypersensitization in the skin by NGF, which can be induced by IL-18 and IL-31, might be significant in DM. The limitation of our study is a small sample size that limited a subgroup analysis. Meanwhile the distinct function of IL-18 and IL-31 in regulating NGF is further required. Nociception and sensory transduction are complicated processes, and the pathogenesis is multifactorial. It is necessary to investigate further to clarify the molecular mechanisms in DM and CLE-associated cutaneous neuropathic symptoms.

## 4. Materials and Methods

### 4.1. Skin Samples

We retrospectively reviewed the skin biopsy specimen database with classic features of DM (*n* = 35) and CLE (*n* = 29) in the Dermatology Department at Kaohsiung Chang Gung Memorial Hospital between July 2014 and May 2021. The diagnosis of DM or CLE was performed by an independent broad-certificated rheumatologist, and the charts review was carried out by the first author. Healthy controls were obtained from the peri-lesioned skin of benign appendage tumors (*n* = 23). This study was approved by the Medical Ethical Committee of the Chang Gung Memorial Hospital. Skin samples were embedded in paraffin sections and were cut by 5 μm slides.

### 4.2. Histological Analysis

All formalin-fixed and paraffin-embedded tissues were obtained from the archives of the Department of Dermatology at Kaohsiung Chang Gung Memorial Hospital, Taiwan. First, 5 µm paraffin sections were deparaffinized by xylene and rehydrated with 100%, 95%, 75%, 50% ethanol, and water. Sections were heated in an autoclave (SX-700, TOMY, Tokyo, Japan) for 5 min in 10 mM citric acid buffer with 0.05% Tween-20. Sections were allowed to cool down. After rinsing with water, sections were blocked with 3% BSA in PBS for 1 h at room temperature. Sections were incubated with primary antibodies overnight at 4 °C in a humid chamber. After being washed with 0.05% Tween-20 PBS, sections were incubated with secondary antibodies for 1 h at room temperature. Nuclei were counterstained with DAPI (D9542, Sigma-Aldrich, St. Louis, MO, USA) and mounted with mounting media. Stained sections were examined using the BX53 microscope equipped with a DP80 camera (Olympus, Tokyo, Japan).

### 4.3. Antibodies

Paraffin-embedded sections were stained with antibodies against human beta3-tubulin, NGF, AREG, and Sema3A. The primary antibodies used in this study were as follows: mouse anti-beta-3 tubulin (Tuj 1) (1:4000 dilution; BIOMOL International LP, Plymouth Meeting, PA, USA), rabbit anti-NGF (1:200 dilution; Chemicon, Temecula, CA, USA), rabbit anti-AREG (1:200 dilution; Bachem, Bubendorf, Switzerland), and rabbit anti-Sema3A (1:200 dilution; Abcam Ltd., Cambridge, UK). Secondary antibodies conjugated with Alexa 488 or Alexa 594 used in this study were obtained from Molecular Probes (1:300 dilution; Eugene, OR, USA).

### 4.4. IENFD and Semi-Quantitative Immunofluorescence Measurements

Tuj-1-immunoreactive nerve fibers above the basement membrane were counted at a magnification of 400 with a light microscope, and at least four fields from at least two different sections of each specimen were collected for analysis. IENFD was quantified on the count of Tuj-1-immunoreactive nerve fibers per millimeter. IENFD is defined as the mean IENF of different fields per millimeter. The number of intra-epidermal nerve fibers per mm2 in the images was counted by hand by two researchers in a blinded manner [42]. All values are presented as mean ± SD of each group. Fluorescence intensity per unit area of epidermal NGF, AREG, and Sema3A was calculated in each group, and statistical analysis was performed.

### 4.5. Cell Culture

Normal human keratinocytes were obtained from adult foreskins. Briefly, skin specimens were washed with phosphate-buffered saline (pH 7.2), cut into small pieces, and harvested in a medium containing 0.25% trypsin (Gibco, Grand Island, NY, USA) overnight at 4 °C. The epidermal sheet was lifted from the dermis with a pair of fine forceps. The epidermal cells were spun down by centrifugation (500× *g*, 10 min) and then dispersed into individual cells by repeated aspiration. The keratinocytes were gently re-suspended in 5 mL of keratinocyte-serum-free medium (Gibco, Grand Island, NY, USA), which contained 25 mg/mL bovine pituitary extract and 5 ng/mL recombinant human epidermal growth factor. Keratinocytes at the third passage were then grown in a keratinocyte-serum-free medium without bovine pituitary extract and recombinant human epidermal growth factor for 24 h before experimentation.

### 4.6. Western Blot Analysis

To verify the signaling pathways, keratinocytes were starved in a serum-free medium followed by cultured with IL-18 at a concentration of 5 ng/mL (Medical & Biological Laboratories Co Ltd., Nagoya, Japan) and IL-31 at a concentration of 300 ng/mL (SRP3091, Sigma-Aldrich, Saint Louis, MO, USA), which referred to a previous study [43], for 6 h, 18 h, and 24 h. Then, cells were lysed with cell lysis buffer (ab152163, Abcam, Cambridge, UK) for 10 min containing protease and phosphatase inhibitor (78443, Thermo Fisher Scientific, Waltham, MA, USA) to perform Western blotting. Cell lysates were quantified with Protein Assay Dye Reagent (5000006, Bio-Rad, Hercules, CA, USA) and SPECTRAMAX-190 (Molecular Devices, San Jose, CA, USA) by detecting the wavelength at 595 nm. Electrophoresis was used to separate 10 or 20 mg cell lysates on 4–12% gradient SDS- PAGE (NP0321BOX, Invitrogen, Waltham, MA, USA) and transferred to a polyvinylidene difluoride membrane (Millipore, Burlington, MA, USA). The blot was blocked with 0.05% Tween-20 phosphate-buffered saline containing 5% skimmed milk or 5% BSA, and then incubated with primary antibody solution at 4 °C overnight. After washing with phosphate-buffered saline with Tween-20, the membrane was incubated with horseradish-peroxidase-conjugated secondary antibody for 1 h at room temperature. Signals were detected with ECL Western Blotting Detection (RPN2106, GE Health, Chicago, IL, USA) and Sygene PXi gel imaging system (Syngene, Cambridge, UK). Antibodies used for Western blotting were anti-NGF rabbit antibody (St John’s Laboratory Ltd., London, UK), and anti-GAPDH rabbit antibody (Sigma-Aldrich, Saint Louis, MO, USA).

### 4.7. Statistical Analyses

One-way ANOVA with Kruskal–Wallis test and multiple comparison with Dunn’s test for three groups comparison, and Student’s *t* test for two groups were used for statistical analyses.

## Figures and Tables

**Figure 1 ijms-23-09030-f001:**
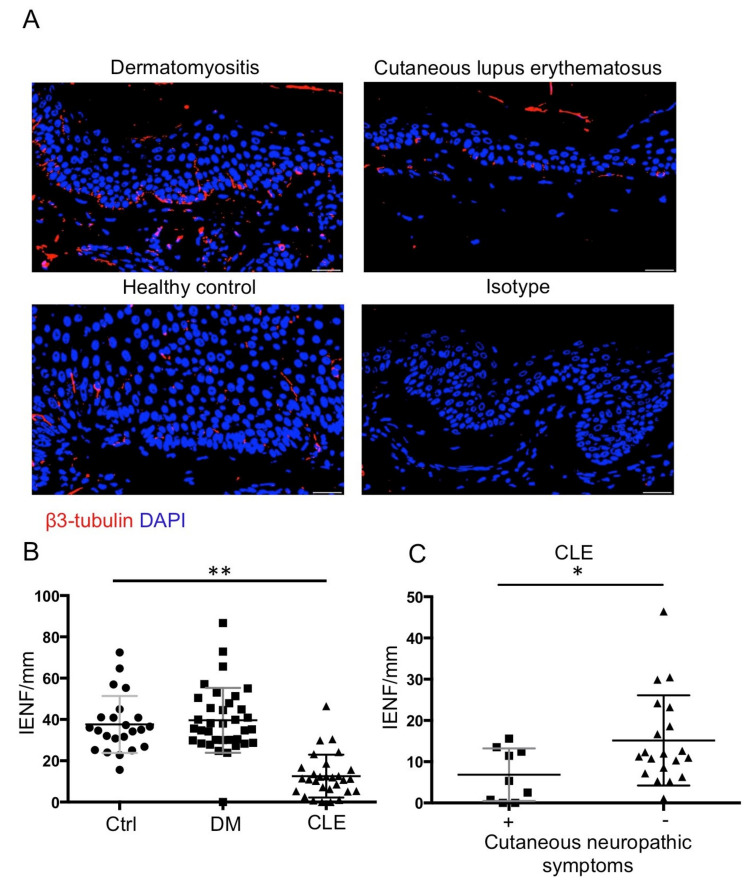
Unchanged IENFD in patients with DM and reduced IENFD in patients with CLE (**A**) Representative images of β3 tubulin (red fluorescence) staining in the paraffinized skin sections. Isotype controls that were stained with non-immune mouse IgG showed no signals in the skin. All sections were counterstained with DAPI (blue nuclear signal). Original magnification: ×400. Bar = 20 μm. (**B**) Quantitative analysis reveals a significant decrease in IENF in patients with CLE (*n* = 29) compared with the healthy controls (*n* = 23) and patients with DM (*n* = 35). (**C**) Quantitative analysis reveals a significant reduction in IENF in CLE patients with cutaneous symptoms (*n* = 9) compared with the asymptomatic group (*n* = 20). All data are presented as the mean ± S.D., * *p* < 0.05 ** *p* < 0.01, one-way ANOVA test for three groups comparison, and Student’s *t*-test for two groups comparison.

**Figure 2 ijms-23-09030-f002:**
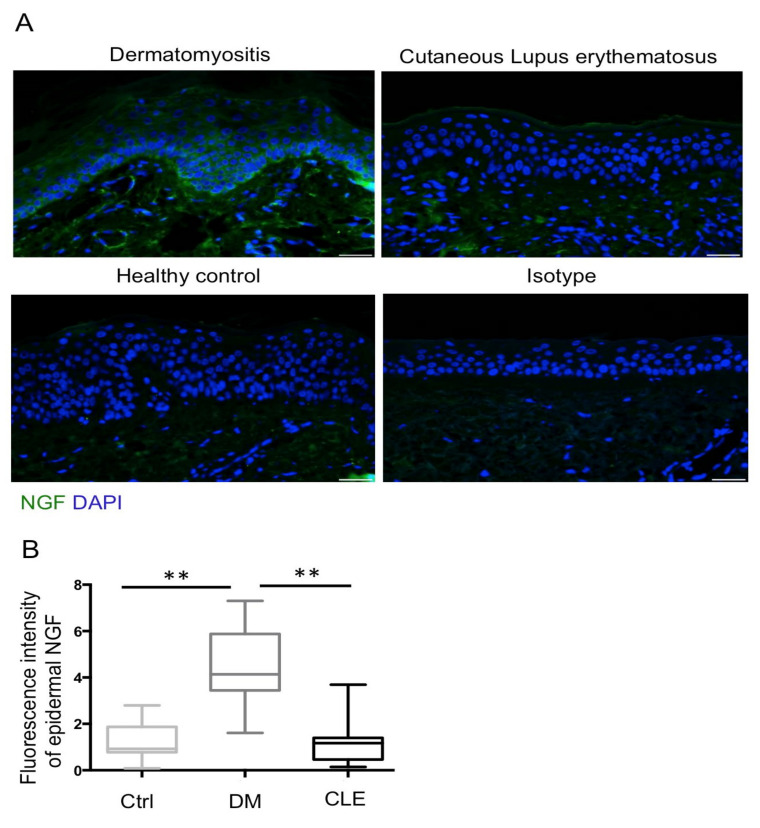
Increased expression of NGF immunofluorescence intensity in the epidermis of patients with DM. (**A**) Representative images of NGF (green fluorescence) staining in paraffinized skin sections. All sections were counterstained with DAPI (blue). Isotype controls were incubated with isotype rabbit IgG. Original magnification: ×400. Bar = 20 μm. (**B**) Semi-quantitative analysis reveals a significant increase in NGF fluorescence intensity in patients with DM (*n* = 35) compared with the healthy controls (*n* = 23) and patients with CLE (*n* = 29). All data are presented as the mean ± S.D., ** *p* < 0.01, one-way ANOVA test.

**Figure 3 ijms-23-09030-f003:**
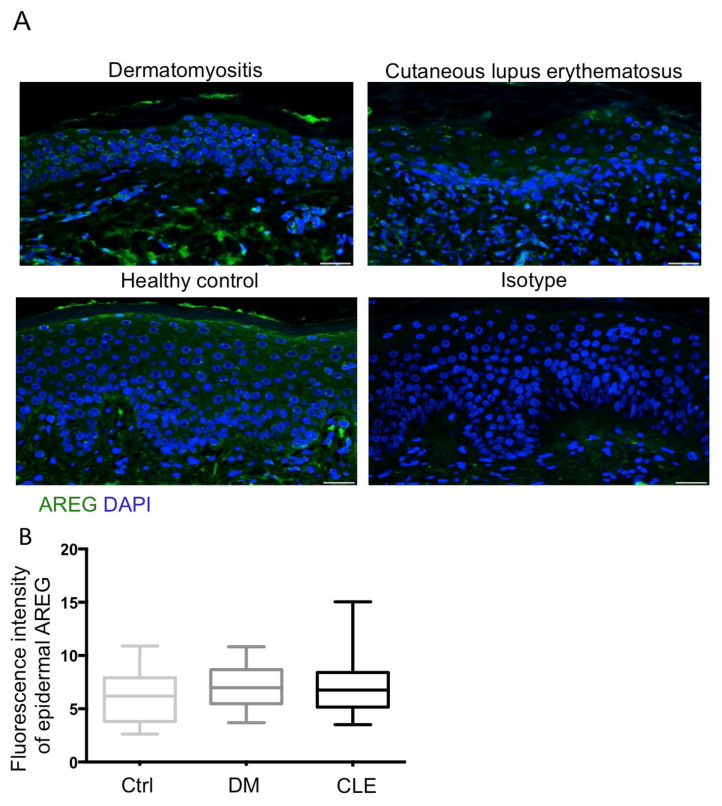
No difference in AREG immunofluorescence intensity between patients with DM and CLE. (**A**) Representative images of AREG (green fluorescence) staining in paraffinized skin sections. All sections were counterstained with DAPI (blue). Isotype controls were incubated with isotype rabbit IgG. Original magnification: ×400. Bar = 20 μm. (**B**) Semi-quantitative analysis reveals comparable AREG fluorescence intensity in patients with DM (*n* = 35), and patients with CLE (*n* = 29) compared with the healthy controls (*n* = 23). All data are presented as the mean ± S.D.

**Figure 4 ijms-23-09030-f004:**
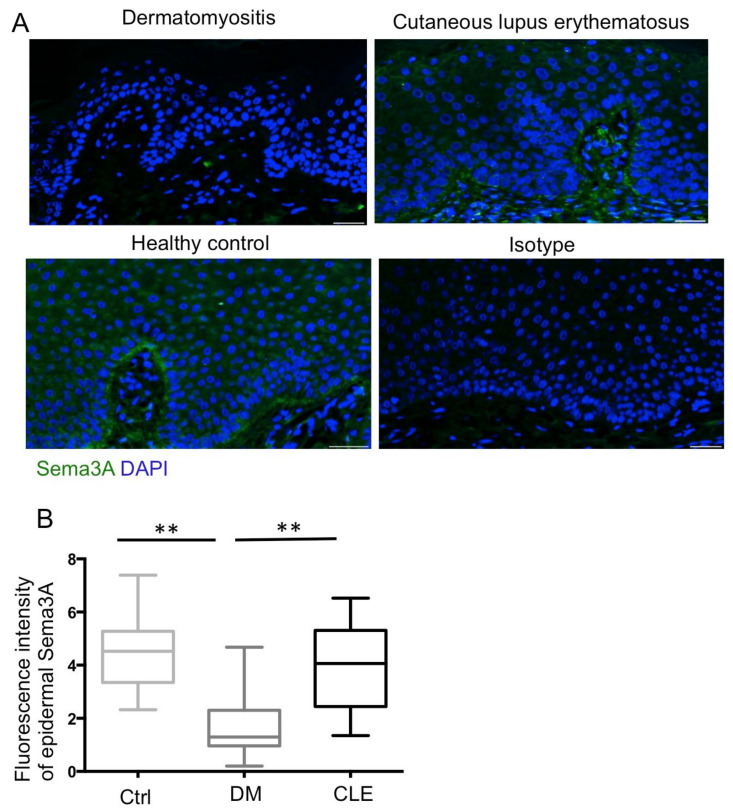
Decreased expression of Sema3A immunofluorescence intensity in the epidermis of patients with DM. (**A**) Representative images of Sema3A (green fluorescence) staining in paraffinized skin sections. All sections were counterstained with DAPI (blue). Isotype controls were incubated with isotype rabbit IgG. Original magnification: ×400. Bar = 20 μm. (**B**) Semi-quantitative analysis revealed a significant decrease in Sema3A fluorescence intensity in patients with DM (*n* = 35) compared with the healthy controls (*n* = 23), and patients with CLE (*n* = 29). All data are presented as the mean ± S.D., ** *p* < 0.01, one-way ANOVA test.

**Figure 5 ijms-23-09030-f005:**
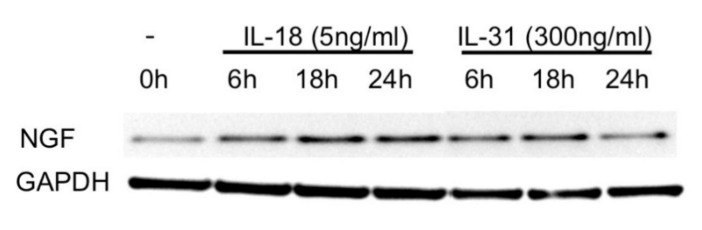
IL-18 and IL-31 induce NGF expression in the keratinocytes. Cultured keratinocytes were treated with IL-18 at 5 ng/mL and IL-31 at 300 ng/mL for the indicated hours. The results were repeated three times with the representative blot shown.

**Table 1 ijms-23-09030-t001:** IENFD in patients with DM and CLE, and healthy controls in different anatomic locations.

	Healthy Control (*n* = 23)	DM (*n* = 35)	CLE (*n* = 29)
Face IENF (/mm)	47.1 +/− 21.3 (*n* = 6)	45.7 +/− 21.5 (*n* = 5)	11.5 +/− 7.3 * (*n* = 19)
Trunk IENF (/mm)	34.2 +/− 9.2 (*n* = 13)	37.2 +/− 9.9 (*n* = 10)	9.5 +/− 12.2 * (*n* = 4)
Limbs IENF (/mm)	34.7 +/− 7.7 (*n* = 4)	42.6 +/− 12.9 (*n* = 20)	9.9 +/− 6.1 * (*n* = 6)

All values are presented as mean ± SD of each group. * *p* < 0.01 compared with healthy controls. IENFD—intraepidermal nerve fiber density; DM—dermatomyositis; CLE—cutaneous lupus erythematosus.

## Data Availability

The data presented in this study are available in the article or the Appendix A.

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
