# Peer review of "Increased Epidermal Nerve Growth Factor without Small-Fiber Neuropathy in Dermatomyositis"

_ijms, 2022, doi:10.3390/ijms23169030_

Round 1

Reviewer 1 Report

dear authors,

This article concerned an interesting study on biological tissue, looking for different concentrations of inflammatory cytokines in DM and CLE conditions, with particular link to SFN. Methodological quality was good, and conduction conforms to literature statement. Below are my comments to improve the manuscript:

- revise the order of the chapters (methods)

- clearly specify that these are preliminary pre-clinical results

- revise the abstract: information should be better clarified with respect to the purpose of the study

- increase the discussion with more in-depth references

Reviewer 2 Report

In this study, authors investigated whether SNF with reduction of IENFD is associated with DM and CLE. The manuscript is carefully written and the reviewers generally agree with the authors' discussion.

I have only one concern. There are many subtypes of morphological skin rash in SLE and DM.

The manuscript uses skin samples from SLE and DM patients, but it is not clear from which skin rash the biopsy was performed. There are subtypes of rash with strong pruritus and some rashes do not pruritus. Even with the same disease, there are variations in the subtypes of the rash, so I am concerned about drawing conclusions lumped in single disease.

Please provide supplementary materials, data that can be linked to all the patient backgrounds used (age, gender, type of rash), in addition to whose samples were used for the Immunofluorescent images in the figures.
